# Vitamin D Deficiency: Consequence or Cause of Obesity?

**DOI:** 10.3390/medicina55090541

**Published:** 2019-08-28

**Authors:** Luka Vranić, Ivana Mikolašević, Sandra Milić

**Affiliations:** Department of Gastroenterology and Hepatology, University Hospital Centre Rijeka, School of Medicine, University of Rijeka, 51000 Rijeka, Croatia

**Keywords:** vitamin D, obesity, deficiency

## Abstract

Obesity is defined as an excess amount of body fat and represents a significant health problem worldwide. High prevalence of vitamin D (VD) deficiency in obese subjects is a well-documented finding, most probably due to volumetric dilution into the greater volumes of fat, serum, liver, and muscle, even though other mechanisms could not completely be excluded, as they may contribute concurrently. Low VD could not yet be excluded as a cause of obesity, due to its still incompletely explored effects through VD receptors found in adipose tissue (AT). VD deficiency in obese people does not seem to have consequences for bone tissue, but may affect other organs, even though studies have shown inconsistent results and VD supplementation has not yet been clearly shown to benefit the dysmetabolic state. Hence, more studies are needed to determine the actual role of VD deficiency in development of those disorders. Thus, targeting lifestyle through healthy diet and exercise should be the first treatment option that will affect both obesity-related dysmetabolic state and vitamin D deficiency, killing two birds with one stone. However, VD supplementation remains a treatment option in individuals with residual VD deficiency after weight loss.

## 1. Introduction

Vitamin D (VD) is essential for the maintenance of bone tissue, as well as for homeostasis of the minerals calcium and phosphorus. Its receptors have been found all over the human body indicating multiple functions. Since body VD is mainly result of endogenous synthesis it is nowadays often considered as a hormone more than vitamin in the narrow sense. The active form of VD, 1,25-dihydroxyvitamin D (1,25(OH)2D), not only stimulates calcium absorption, osteoclastic bone resorption, and osteoblast function and decreases PTH (parathyroid hormone) secretion but also has extraskeletal functions such as decreasing collagen type 1 production, enhancing muscular function, and stimulating cell differentiation, insulin secretion, and the immune system [1]. A severe lack of VD causes osteomalacia in adults and rickets in children, while a less severe deficiency leads to increased bone turnover and consequently a greater risk of bone fractures. Lately, VD deficiency, defined as a serum concentration of 25-hydroxyvitamin D (25(OH)D) below 50 nmol^−1^ or 20 ng mL^−1^ [2,3], has been associated with numerous disorders, such as cardiovascular diseases, arterial hypertension, dyslipidemia, type 2 diabetes, cancer, multiple sclerosis, depression, dementia, psychiatric diseases, and others [4]. Hence, today, VD and its unclear role in the pathogenesis and progression of those disorders represent a trend in scientific research.

Obesity is defined as an excess amount of body fat and represents a significant health problem world-wide. The association between VD deficiency and obesity as well as with obesity-related diseases has been confirmed by numerous studies, but the presence of a causal relationship is still unclear. There are many possible explanations regarding the inverse relationship between increased adiposity, particularly abdominal obesity, and low plasma VD concentrations, as discussed below, but so far none of these hypotheses can completely elucidate this relation. Therefore, it is possible that several mechanisms have an impact on the interaction between VD, obesity, and related diseases. Hence, more studies are needed to investigate each one, not only to investigate the causal relationship, but also to identify and establish VD supplementation treatment that could possibly positively affect adiposity and related disorders. However, so far, studies have shown inconsistent results regarding the clinical implication of VD supplementation, which raises the following main question. Is VD deficiency only a coincidental finding in obese subjects or could it actually have an important role in the development and progression of obesity and chronic illnesses?

## 2. Clinical Consequences of Lower 25(OH)D

A low plasma level of VD usually leads to impaired calcium absorption in the bowel and a lower plasma calcium level, which consequently leads to enhanced bone turnover and impaired bone mineral density (BMD). However, obese subjects have a greater BMD then lean people, as well as greater cortical thickness and cortical tissue mineral density. Hence, obesity has a positive impact on peak bone mass acquisition [5]. The lack of VD deficiency consequences on bones may indicate that, in fact, although the plasma 25OHD level is usually lower, obese people are actually not VD insufficient. The greater VD reservoir in obese people can possibly act as a permanent supply that consequently maintains bone turnover. On the other hand, greater skeletal loading and actions of hormones such as estrogen, leptin, and adiponectin could compensate for an eventual VD deficiency, leading to a greater BMD [6].

Low VD is also associated with poor health behaviors. Recently, its deficiency has been associated with a large number of disorders such as metabolic syndrome, cancers, and autoimmune, psychiatric, and neurodegenerative diseases, but the causative role of VD deficiency in many of these conditions remains unclear [7]. Since VD deficiency is related to visceral adiposity; it could be used as a biomarker of a visceral adiposity-related dysmetabolic state (cardiovascular diseases, type 2 diabetes, dyslipidemia, arterial hypertension) (Figure 1). Nevertheless, its independent role in the development and progression of these diseases cannot yet be excluded, due to changes in the expression of genes regulated with VD receptors. Moreover, as stated below, in VD deficient subjects, the lack of possible anti-inflammatory effects on chronic low-grade inflammation could lead to an enhanced risk of obesity-related metabolic disorders.

## 3. Causes of Low 25-Hydroxyvitamin D in Obese Patients

Nowadays, volumetric dilution of VD is the most probable mechanism of the inverse relationship between vitamin D serum levels and BMI. Even though obese and lean subjects have similar amounts of VD, in overweight people, VD is distributed into a larger volume, making serum concentrations lower. Namely, 25(OH)D is distributed dominantly into the serum, muscle, fat, and liver—compartments that are increased in obesity [6]. The explanation for this could be found in the fact that seasonal variation has a large impact on the difference in serum 25(OH)D concentration between normal weight and obese groups. According to Bolland et al. [8], the difference is greater in summer, because the increase in 25(OH)D serum levels due to sunlight exposure is less in obese people in comparison to normal weight groups because of distribution to compartments other than serum. Moreover, another study reported a smaller 25(OH)D rise in serum in obese than in normal weight people in response to VD supplementation [9]. Accordingly, Drincic et al. [10] suggest that VD supplementation needs to be adjusted for body size in order to erase the serum rise difference between obese and lean subjects. In addition, in their study, the 25(OH)D response after VD supplementation was approximately 30% lower in the obese group. Carelli et al. [11] measured the concentration of VD in the plasma and omental and subcutaneous tissue in both obese individuals and a control group, and they found that the relationship between plasma VD and the VD concentrations in the subcutaneous and omental fat compartments was similar between both groups and that the pattern of distribution of VD between those two fat tissues was also similar, indicating that adipose tissue (AT) does act as a reservoir for VD. However, if volumetric dilution is the main cause of low 25(OH)D in obese people, this suggests that weight loss would, consequently, increase VD serum levels. Nevertheless, weight loss studies show inconsistent results. Some report increased 25OHD serum levels [12,13]; while, on the other hand, others show insignificant increases in serum levels. Mason et al. [14] conducted a weight loss study on a large number of obese postmenopausal participants (N = 398) who underwent a 12 month long weight loss plan involving calorie restriction and exercise and showed that weight loss increased 25(OH)D serum levels insignificantly. However, in the group of women who lost more than 15% of their body weight, 25(OH)D significantly increased (7.7 ng/mL), which suggests that there may be a threshold of weight loss that can positively alter the VD serum concentration.

On the other hand, VD sequestration in adipose tissue (AT) hypothesis was introduced by Wortsman et al. [15] who demonstrated that even though dermal synthesis of VD does not differ between two groups, obese subjects have lower increments in plasma levels of 25(OH)D after sunlight exposure and oral supplementation of VD compared to normal weight subjects. They suggested that VD, as a fat-soluble vitamin, is accumulated and retained in AT, which leads to lower plasma levels of VD in people with a large amount of AT. This sequestration hypothesis was the basis for the volumetric dilution hypothesis that is discussed previously. However, in comparison to volumetric dilution, sequestration of the prohormones ergocalciferol (VD2) and cholecalciferol (VD3) refers not only to their hydrophobic nature and tendency to dissolve in AT but also refers to their inability to regress into the circulation as a substrate for liver 25-hydroxylase, which converts these prohormones to 25(OH)D once they are stored [16].

Another possible mechanism for lower 25(OH)D is impaired hepatic 25-hydroxylation. Targher et al. [17] reported that 25-hydroxylation is impaired in patients with non-alcoholic fatty liver disease (NAFLD), a condition that is very common in obesity. In addition, they found that decreased 25(OH)D serum concentrations are closely related to the severity of histologically proven liver steatosis, inflammation, and necrosis. However, recently NAFLD has become the most common form of chronic liver disease and, consequently, the leading cause of liver cirrhosis. Since it is strongly associated with obesity and metabolic syndrome, large randomized, placebo-controlled trials are needed to confirm the relation between this condition and low 25OHD and to evaluate the possible positive effect of VD supplementation [4,18].

Furthermore, one study was performed in order to elucidate whether low serum levels of VD in obese people could be due altered VD metabolism in AT. They showed that there is a difference in gene expression in VD metabolizing enzymes between normal weight and obese people, which suggests that AT could be involved in the metabolism of VD and does not only passively store fat-soluble nutrients. In fact, they found a 71% (*p* < 0.0001) decrease in the expression of the cytochrome P450 2J2 gene, which codes for the enzyme 25-hydroxylase, and a 49% (*p* < 0.05) decrease in the expression of cytochrome P450 27B1, which codes for the enzyme 1a-hydroxylase in the subcutaneous AT of the obese group compared to lean subjects [12,16]. Since these enzymes play roles in particular steps of hydroxylation or in the conversion of prohormones into bioactive form, the observation of a decrease in gene coding for them would implicate a deficit of the VD bioactive form and a reduced effect in the body. Moreover, the difference in the expression of cytochrome P450 24A1, which codes for the enzyme responsible for inactivation of 1,25(OH)2D (a bioactive form), was not observed between obese and normal weight groups. Nevertheless, after weight loss the expression of this gene was increased by 79%. This upregulation of VD inactivating genes after weight loss and downregulation of bioactivating genes in obesity may suggest involvement of AT in VD metabolism. In addition, these contrary findings represent a puzzling phenomenon and require further research as to whether gene expression differences lead to altered VD metabolism, and if that proves true, whether the local AT metabolism of VD could actually impact the circulating levels of 25(OH)D. Moreover, the lack of bioactivating enzymes in AT indicates that volumetric dilution is not the only reason why obese people need higher doses of VD in order to achieve target concentrations [16].

Considering that skin synthesis of cholecalciferol through ultraviolet B (UVB) radiation from sun exposure is the main source of VD (80–90%) [3], the low concentration of 25(OH)D, the circulating form of VD that is usually measured in trials due to its relatively long half-life, observed in obese individuals could be explained by less sunlight exposure due to their lower mobility and participation in outdoor activities and different clothing habits than normal weight people [6,16]. Two UK studies, on the other hand, affirmed that sunlight exposure between normal weight people and obese people does not vary [19,20]. Another study reported that, when exposed to UVB, cutaneous synthesis of VD is similar among people with different body mass index (BMI) values [15] (Figure 2). Nevertheless, due to geographic and cultural differences sun exposure could have impact on VD synthesis and consequently serum levels in some groups.

Adverse dietary habits are common in obesity, which could lead to a lower intake of VD. Since dietary sources are minor contributor to total VD intake in humans, this explanation is less probable [21]. Moreover, Walsh et al. [6,19] reported that the dietary intake of VD in the UK population does not differ between obese and normal weight people. Thus, diet is most probably irrelevant factor even though its minor contribution to low VD levels could not be completely excluded.

Lately, there have been some studies and experimental data that support the hypothesis that VD could be involved in the pathogenesis of obesity, rather than just being a consequence. Some experimental data suggest that an increased level of parathyroid hormone, due to VD deficiency, promotes lipogenesis by greater calcium inflow in adipocytes [3,22]. Another, more probable, hypothesis is that the active form of VD, 1,25(OH)D, inhibits adipogenesis through actions modulated by VD receptors [3,23]. Blumberg et al. [24] demonstrated that in 3T3-L1 preadipocytes, in the presence of 1,25(OH)D, VD receptors inhibited differentiation by downregulating the adipocyte promoting transcription factor C/EBPβ. Furthermore, 1,25(OH)D is able to maintain the WNT/β-catenin pathway, which is downregulated during adipogenesis, and thus can inhibit adipogenesis [25]. Therefore, lower VD levels could lead to enhanced differentiation of pre-adipocytes to adipocytes. Moreover, two independent longitudinal studies showed that a lower VD concentration predisposes an individual to obesity and is associated with greater weight gain compared to subjects with weight gain but a higher baseline VD [26,27]. Due to lack of clinical studies, the exact role of VD deficiency in development and predisposition to obesity still remains questionable. It is definitely more probable that VD deficiency is just a consequence due to volumetric dilution and other causes but, since experimental data showed promising results, involvement in pathogenesis could not be excluded. Therefore, more prospective randomized controlled trials are needed, because if involvement is proven, that could possibly establish VD supplementation as a treatment option in preventing obesity, even though VD supplementation trials show inconsistent results, as discussed below.

## 4. Effects of Vitamin D Supplementation

Since a low VD could contribute, or at least is related to dysmetabolic condition and related diseases, as discussed previously, the main question remains: Could VD supplementation therapy have long-term health benefits for visceral adiposity-related disease patients?

Numerous studies have demonstrated conflicting results regarding the management of adiposity-related diseases with VD supplementation. Zittermann et al. [28] demonstrated that VD supplementation does not adversely affect bodyweight, but it could significantly improve several cardiovascular risk markers. Other studies also showed no effect of VD treatment on bodyweight reduction and body composition [29,30,31]. These findings indicate that, even though a low VD concentration is associated with obesity, the association is not bidirectional. 

Studies suggest that an increase in plasma VD is not associated with a significant increase in blood pressure [29,32,33]. VD therapy has shown inconsistent findings in managing adverse lipid profile, namely, higher triglycerides, total, and LDL cholesterol concentrations and a lower HDL concentration. One double blind, placebo controlled randomized clinical trial demonstrated no effect of increased 25(OH)D on the lipid profile in obese patients [29], while another study noted an enhanced impact of VD supplementation on the decrease of serum triglycerides in healthy obese patients during weight loss [28].

Epidemiological, experimental, and clinical studies also suggest a possible key role of chronic low-grade inflammation in the development of metabolic syndrome features, such as type 2 diabetes [34] and cardiovascular diseases [35]. Two studies on adipose tissue cultures demonstrated anti-inflammatory effects of 1,25(OH)2D under experimental conditions [36,37]. If anti-inflammatory effects could be proven in vivo, this would suggest that either low VD is a cause and/or that increasing the plasma VD concentration could reduce low-grade inflammation, as seen as a decrease in circulatory inflammatory markers. In obese subjects, during a weight loss trial, year-long daily supplementation of 3332 IU VD augmented the decrease in TNFα, but decreases in CRP and IL-6 were not observed compared to a placebo [28], and in patients with congestive heart failure, ingestion of 2000 UI of VD daily for 9 months decreased plasma TNFα, increased IL-10, and did not affect CRP [38]. Conversely, Pittas et al. [39] demonstrated that daily treatment of 700 IU VD and 500 mg calcium for 3 years did not affect circulating levels of cytokines. Moreover, Wamberg et al. [36] reported that a daily VD dose of 7000UI for 26 weeks affected neither the expression of inflammatory markers in AT, nor concentrations of circulating cytokines. Since clinical studies, in opposite to experimental studies, did not show decreases in inflammatory markers levels, or at least results were inconsistent, VD supplementation is not recommended for that matter, even though more studies are needed. 

Insulin resistance is a commonly seen condition in obese people and predisposes individuals to the development of type 2 diabetes. Inconsistent findings have been reported regarding the effect of VD supplementation on insulin resistance. Several cross-sectional studies have shown that VD deficiency is associated with hyperglycemia [40,41], hyperinsulinemia [42], impaired β-cell function, and insulin resistance [43]. Von Hurst et al. [44] found decreased HOMA-IR after VD treatment in overweight and insulin resistant women, but also found no effect on the fasting glucose level, lipid profile, and inflammatory markers. However, other trials have demonstrated a lack of effect of VD treatment on insulin resistance. Wamberg et al. [29] found no effect on insulin resistance after 26 weeks of treatment with 7000 UI of cholecalciferol, while another prospective observational study actually reported an association between low VD and the development of type 2 diabetes, which became insignificant after adjustment for BMI [45]. In VD deficient men, short term treatment with high doses of cholecalciferol did not improve insulin resistance measured with HOMA-IR [46]. These contrasting findings suggest that further studies are required in order to establish whether VD supplementation is a possible treatment for people who are predisposed to type 2 diabetes. 

Wamberg et al. [29] also explored the effects of VD supplementation on ectopic fat accumulation by measuring the hepatic lipid content with MR spectroscopy and reported that the concentration of intrahepatic lipids strongly correlates with insulin resistance and consequently with the development of type 2 diabetes and cardiovascular incidents. These findings are consistent with another study that found a correlation between the severity of NAFLD based on histopathological analysis and a low plasma VD concentration [17]. Nevertheless, in their study, Wamberg et al. did not find a decrease in intrahepatic fat accumulation after VD treatment. Based on their findings they suggest that steatotic liver, a common feature in obesity, is associated with a decreased capacity for the hydroxylation of prohormones into 25(OH)D.

In Table 1 are studies that investigated the association between obesity and VD.

## 5. Conclusions

The high prevalence of VD deficiency in obese subjects is a well-documented finding that is most probably due to volumetric dilution into the greater volumes of fat, serum, liver, and muscle present in obese people. However, other mechanisms cannot completely be excluded, as they may contribute concurrently. A low VD concentration cannot yet be excluded as a cause of obesity due to the incompletely explored effects of VD receptors found in AT. VD deficiency in obese people does not seem to have consequences for bone tissue, but may affect other organs, even though studies have shown inconsistent results and VD supplementation has not yet been clearly shown to benefit the dysmetabolic state. Hence, more studies are needed to determine the actual role of VD deficiency in the development of these disorders and the effect of VD supplementation. Thus, treatment with VD still remains controversial. Nevertheless, when it is required, due to volumetric dilution in obese patients, higher doses of vitamin D are needed to achieve the same serum concentration compared to lean subjects [47]. Once achieved, the maintenance dose should not differ between the normal weight and obese groups. However, weight loss is currently the only actually proven treatment that leads to improvement in a number of disorders, including VD deficiency. Adiposity loss, especially the loss of visceral fat tissue, has been strongly associated with better health outcomes contributing to the normalization of insulin resistance, adverse lipid profiles, and arterial hypertension, which are all features of metabolic syndrome [16]. Thus, targeting lifestyle improvements through the promotion of a healthy diet and exercise should be the first treatment option, as it will affect both the obesity-related dysmetabolic state and vitamin D deficiency, killing two birds with one stone. However, VD supplementation remains a treatment option for individuals with residual VD deficiency after weight loss.

## Figures and Tables

**Figure 1 medicina-55-00541-f001:**
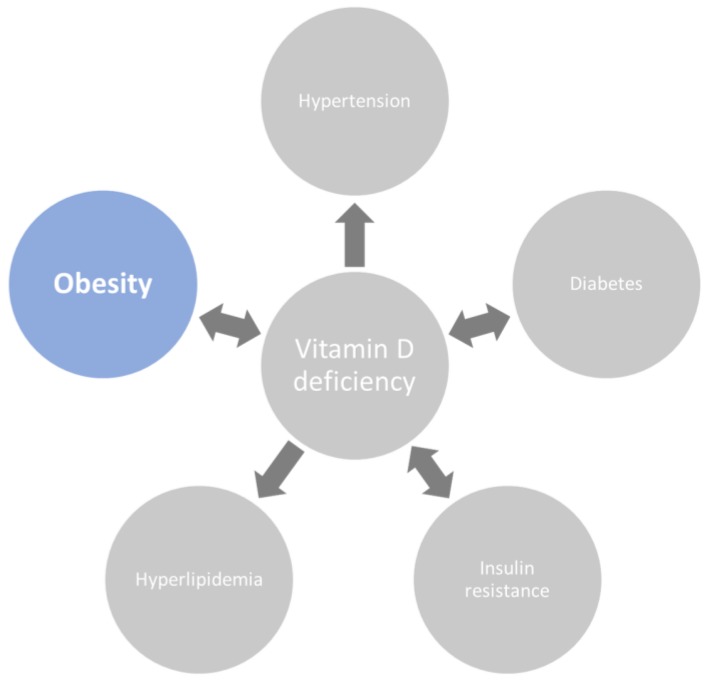
Vitamin D deficiency is strongly related to obesity as well as to other components of adiposity-related dysmetabolic state.

**Figure 2 medicina-55-00541-f002:**
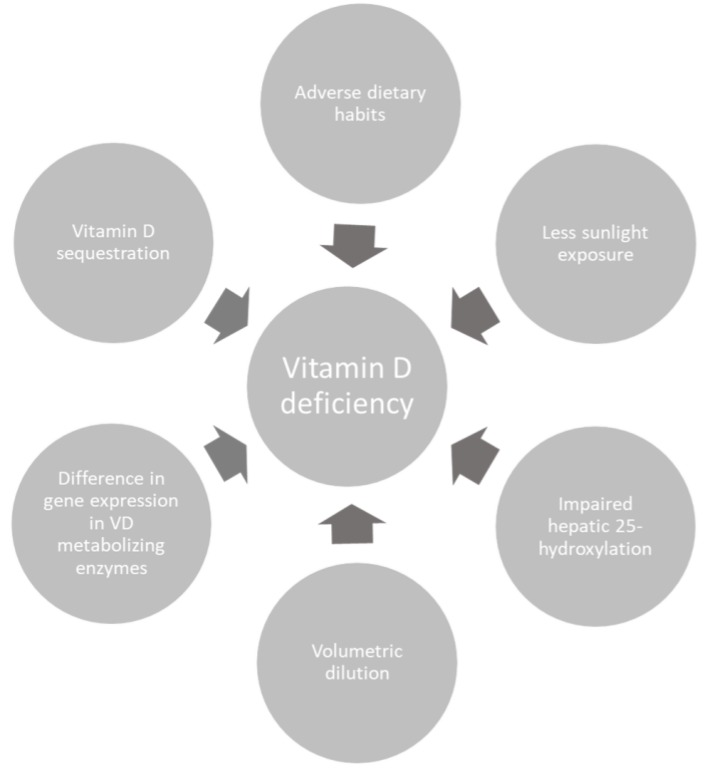
Possible causes of vitamin D deficiency.

**Table 1 medicina-55-00541-t001:** Studies that investigated the connection between obesity and vitamin D.

Ref	Study Design	Study Size	Major Findings
Evans et al. [5]	Cross-sectional case-control study	200	Obesity has a positive impact on peak bone mass acquisition and also obese subjects have greater cortical thickness and cortical tissue mineral density.
Bolland et al. [8]	Cross-sectional study	1984	Vitamin D serum levels show seasonal variations.
Gallagher et al. [9]	Randomized, double-blind placebo controlled study	163	Obese subjects respond with smaller 25(OH)D rise in serum after VD supplementation in comparison to normal weight group.
Drincic et al. [10]	Randomized, single-blind study	67	25(OH)D response after VD supplementation is 30% lower in the obese group. VD supplementation should be adjusted according for body size.
Wamberg et al. [12]	Cross-sectional study	40	Due to different expression of liver enzymes between obese and normal weight groups, adipose tissue can metabolize VD locally which can be altered after weight loss.
Rock et al. [13]	Prospective randomized clinical trial	383	Weight loss increases 25(OH)D serum concentration
Mason et al. [14]	Prospective randomized controlled trial	439	Weight loss of 15% of body weight and above increases 25(OH)D significantly, otherwise weight loss does not impact on serum 25(OH)D
Walsh et al. [19]	Cross-sectional observational study	223	Total and free 25(OH)D and 1,25(OH)2D are lower at higher BMI does not impact bone structure and health.
Macdonald et al. [20]	Prospective observational cohort study	314	Vitamin D deficiency is not related to reduced sun exposure in obese woman. Vitamin D serum concentrations seasonally changes.
LeBlanc et al. [26]	Prospective longitudinal study	9704	Vitamin D deficiency predisposes for obesity, higher doses of VD are related to lower weight gain
Mai et al. [27]	Cross-sectional and prospective cohort study	25,616	Low plasma 25(OH)D level (less than 50 nmol/L) was related to higher BMI and waist circumference.
Zittermann et al. [28]	Randomized, double-blind placebo-controlled study	200	VD supplementation has positive impact on several cardiovascular disease risk markers in obese, but does not adversely affect weight loss
Wamberg et al. [29]	Randomized, double-blind placebo-controlled study	52	VD supplementation has no effect on obesity-related complication nor on body weight reduction.
Kampmann et al. [30]	Randomized, double-blind placebo-controlled study	16	VD supplementation does not improve insulin resistance, blood pressure, inflammation or HbA1c, but might increase insulin secretion.
Mason et al. [31]	Randomized, double-blind placebo-controlled study	218	VD supplementation does not reduce body weight.
Wamberg et al. [36]	Randomized, double-blind placebo-controlled study	40	Inflammatory cytokines and inflammatory markers expression in adipose tissue were not reduced after VD supplementation, nevertheless it had significant inflammatory effects in AT in vitro.
Schleithoff et al. [38]	Randomized, double-blind placebo-controlled study	123	Improvement of VD status decreased plasma proinflammatory cytokines in patients with congestive heart failure.
Pittas et al. [39]	Randomized, double-blind placebo-controlled study	314	VD and calcium supplementation did not affect circulating levels of cytokines but attenuate increases in glycemia and insulin resistance in nondiabetic subjects.
Baynes et al. [42]	Prospective population-based cross-sectional study	142	VD hypovitaminosis is associated to hyperglycemia.
van Hurst et al. [44]	Randomized, double-blind placebo-controlled study	81	VD supplementation improves insulin sensitivity, but has no effect on insulin secretion.
Drincic et al. [47]	Cross-sectional study	686	Body weight and VD plasma levels are conversely associated due to volumetric dilution.

Vitamin D (VD).

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
