# Peer review of "Vitamin D Deficiency: Consequence or Cause of Obesity?"

_medicina, 2019, doi:10.3390/medicina55090541_

Round 1
Reviewer 1 Report
Nice review on Vit D and obesity. All the questions can be positively answered; the paper summarizes the state of the art in regard to obesity and Vit D!
I recommend to accept!
Author Response
Dear reviewer,
thank you very much for a recommendation. I am glad that you think that our article summarizes all important information regarding obesity and vitamin D.
Best regards,
Authors
Reviewer 2 Report
The content of this manuscript represents an important discussion for the field that has not been well represented as a review previously. Thus, it is an important contribution to the field. There is much information contained, although a few more references to vitamin D metabolism in different tissues (including adipose) would contribute to its strength. However, the manuscript is disorganized, both in the overall structure and within just about every paragraph. There are many cumbersome or long and complex sentences, as well as sentences were words are (I think) misused and thus make the sentence hard to follow.
Some examples are that it is not clear why the basis for the dilution hypothesis is not presented before the alternatives, issues or other interpretations. It is also not clear why the consequences of vitamin D deficiency are after the hypothesis. It is possible the current structure will work if the rest if written in a clear and logical fashion. The arguments simply aren’t clearly stated, and many paragraphs contain multiple points and sometimes no conclusion sentence, so it is difficult to determine the intent of the paragraph and to grasp the argument.
One good example is in the abstract as follows; the wording is cumbersome, the sentence too long and complex. I am not really sure what the authors are trying to say without dissecting the sentence myself and trying to think about it. Shouldn’t be so difficult for the reader, and someone not familiar with the field would not understand it:
‘Thus, targeting lifestyle through healthy diet and exercise should be the first treatment option that will affect both obesity-related dysmetabolic state and vitamin D deficiency, killing both birds with one stone, while VD supplementation remains still a treatment option in individuals with residual VD deficiency after weight loss.’
There are also a number of places where the writing assumes a conclusion, or states a correlation is proof of causation. For example: ‘and to eventually establish VD supplementation as a treatment option’ would only be the case if there is a strong argument for this, but it has not yet been presented.
Other points include that the definition of a hormone is both that it is made in the body and acts distally. This is not represented well in the second sentence of the introduction.
1,25(OH)2D is not defined when first presented (third sentence introduction) and neither is 25(OH)D. The full name should be spelled out the first time with the definition in parentheses.
‘uncleared role’ is not correct English.
‘progress of those disorders represents a trend in scientific research.’ Is not clear nor grammatically correct.
‘The association between VD deficiency and obesity and low VD and obesity-related diseases has been confirmed by numerous studies, but the presence of a causal relationship is still unclear.’ Three ‘and’s makes this too complicated to understand.
Section 2 paragraph one lacks a conclusion sentence, and with the writing in the rest of the paragraph leaves the intent of this paragraph unclear.
‘However, since NAFLD has recently become the most common form of chronic liver disease and, consequently, the most common cause of liver cirrhosis, and because it is strongly associated with obesity and metabolic syndrome, large randomized, placebo-controlled trials are needed to confirm the relation between this condition and low 25OHD and to evaluate the possible positive effect of VD supplementation [4,12] . ‘ This is a long and complex sentence and very difficult to understand.
‘his upregulation of VD inactivating genes after weight loss and downregulation of bioactivating genes in obesity proves involvement of AT in VD metabolism.’ This does not ‘prove’ although it may suggest. Observation does not equal causation.
‘This is also because the metabolic resistance to 25(OH)D increases [5]. ‘ I’m not sure what this means; the argument in the paragraph does not lead me to this conclusion (if I understand what resistance to 25(OH)D means).
These are some examples of issues in the writing, with some depth to help the authors understand and perhaps correct these issues.
Author Response
Dear reviewer,
In attachment you will find enclosed responses on your comments and thoroughly described step by step what we have changed in order to meet your expectations. Thank you for the nice recommendations to make this article better.
Best regards,
Authors

Round 2
Reviewer 2 Report
The topic is timely and important. The authors present a well thought out and supported position that is contrary to the prevailing opinion, thus making a significant contribution to the field.
There remain issues with using English words incorrectly, leading to awkward phrasing in some places. There are other grammatical errors as well. However, the manuscript is significantly improved in use of language and organization.